# Improved Reconstruction of MR Scanned Images by Using a Dictionary Learning Scheme

**DOI:** 10.3390/s19081918

**Published:** 2019-04-23

**Authors:** Shahid Ikram, Jawad Ali Shah, Syed Zubair, Ijaz Mansoor Qureshi, Muhammad Bilal

**Affiliations:** 1Department of Electrical Engineering, International Islamic University Islamabad, Islamabad 44000, Pakistan; Shahid.ikram@iiu.edu.pk (S.I.); szubair@iiu.edu.pk (S.Z.); m.bilal@iiu.edu.pk (M.B.); 2Department of Electrical Engineering, UniKL BMI, Kuala Lumpur 53100, Malaysia; 3Department of Electrical Engineering, Air University Islamabad, Islamabad 44000, Pakistan; imqureshi@mail.au.edu.pk

**Keywords:** compressed sensing (CS), dictionary learning, magnetic resonance imaging (MRI), focal underdetermined system solver (FOCUSS), simultaneous code word optimization (SimCO), and dictionary learning based MRI (DLMRI)

## Abstract

The application of compressed sensing (CS) to biomedical imaging is sensational since it permits a rationally accurate reconstruction of images by exploiting the image sparsity. The quality of CS reconstruction methods largely depends on the use of various sparsifying transforms, such as wavelets, curvelets or total variation (TV), to recover MR images. As per recently developed mathematical concepts of CS, the biomedical images with sparse representation can be recovered from randomly undersampled data, provided that an appropriate nonlinear recovery method is used. Due to high under-sampling, the reconstructed images have noise like artifacts because of aliasing. Reconstruction of images from CS involves two steps, one for dictionary learning and the other for sparse coding. In this novel framework, we choose Simultaneous code word optimization (SimCO) patch-based dictionary learning that updates the atoms simultaneously, whereas Focal underdetermined system solver (FOCUSS) is used for sparse representation because of a soft constraint on sparsity of an image. Combining SimCO and FOCUSS, we propose a new scheme called SiFo. Our proposed alternating reconstruction scheme learns the dictionary, uses it to eliminate aliasing and noise in one stage, and afterwards restores and fills in the *k*-space data in the second stage. Experiments were performed using different sampling schemes with noisy and noiseless cases of both phantom and real brain images. Based on various performance parameters, it has been shown that our designed technique outperforms the conventional techniques, like K-SVD with OMP, used in dictionary learning based MRI (DLMRI) reconstruction.

## 1. Introduction

Magnetic resonance imaging (MRI) is one of the important tools used to generate anatomical images of the body without damaging and harmful radiation. A strong magnetic field and radio waves are used in MRI to produce detailed images of the organs and soft tissues within the body. These days, MRI is considered as the main source of reproducible diagnostic medical information because of its noninvasive technique and accurate visualization of the anatomical skeleton. It is essential to take a higher number of measurements to cover a large dynamic range of tissue parameters in relevant medical applications for reconstruction of the image with clear features.

The main problem in MRI is the acquisition time to get a large number of image samples for a good reconstruction of the image. So a patient has to stay and endure a long time in an MRI machine. The problem of long acquisition time in a machine can be reduced by making development either on the hardware side or the software side. The changes in the software side can be implemented more easily than for the hardware side through efficient algorithms. These algorithms are mainly based on compressed sensing (CS).

The application of CS to biomedical imaging is exciting since it permits an almost exact reconstruction of images from far fewer measurements. For biomedical imaging, CS can increase the imaging speed and consequently decrease the radiation dose. Modern theory of CS [1,2,3,4,5,6,7,8,9,10,11] supports recovering images accurately using fewer measurements than the number of unknowns as required by conventional Nyquist sampling. This is possible, provided the underlying image is sparse in some transform domains. The cost of this improvement is that the reconstruction technique is nonlinear. In recent times, CS theory has been used in MRI [12,13,14,15,16,17] showing good quality reconstructions from a reduced set of measurements. 

The quality of CS reconstruction methods largely depends on the use of various sparsifying transforms such as wavelet or total variation (TV), to recover magnetic resonance imaging (MRI), computed tomography (CT) [18] and other biomedical images from the subsampled data by exploiting the sparsity of these images in a transform domain or dictionary. Non-adaptive global sparsifying transforms are restricted in conventional MR images to 2–4 folds under-sampling [3,4,14,19]. Numerous unwanted artifacts and loss of features were observed during the reconstruction of images with non-adaptive sparsifying transforms (dictionary) like wavelet and TV, etc.

Initially, fixed sparsifying transforms known as non-adaptive sparsifying transform (or dictionary learning) were used to reconstruct the medical images. A lot of work has been conducted to learn adaptive sparsifying transforms (dictionary) which can better sparsify the images because these are trained from the particular class of images [20,21]. Different artifacts and aliasing effects come into play on the edges of reconstructed images, when using under-sampled data from *k*-space. Patch-based sparsity dictionary learning has a tendency to capture the local image features effectively and can have potential to eliminate the aliasing artifact without compromising the resolution. An adaptive patch-based dictionary learnt from a small number of *k*-space samples provides a promising reconstruction.

Dictionary learning involves a two-step process, i.e. the dictionary learning from training data and sparse representation. Saiprasad et al [22] presents DLMRI scheme based on K-SVD for learning sparsifying transform with OMP for sparse coding. There are also some other methods, such as MOD, ILS, RLS and SimCO for dictionary learning and BP [23,24,25,26], LASSO [27], and FOCUSS [28] for sparse coding.

A lot of research work is being carried out on deep learning, particularly in convolutional neural networks (CNNs), for MRI reconstruction [29,30,31,32] but the main bottle neck is the availability of high computation resources and large amount of data. In our proposed work, we conduct dictionary learning using a single image. This single image cannot be used for training a deep learning network. Hence, we restrict our work to a dictionary learning based method.

A well-known algorithm K-SVD is used extensively for dictionary learning with OMP for sparse coding to reconstruct the image. The K-SVD updates the columns sequentially one by one and takes time to update all the atoms. This technique also faces the increase of singular vectors. Although the K-SVD performs well in capturing a reference dictionary, its denoising performance is comparatively limited as shown in our simulation in Section 3. The artifacts at the edges appear due to the hard constraint applied by the orthogonal matching pursuit (OMP). This hard sparseness constraint may not be very useful for a medical imaging application which produces artifacts on a high frequency.

To address the above-mentioned issues, we are using FOCUSS as a sparse coding technique because of its soft constraints, which is considered to be better for sparse representation of medical images. The pruning process of FOCUSS has a tendency to suppress the noise during the reconstruction of image because aliasing artifacts and noises are usually isolated. This is the main reason for our proposed scheme, which performs well particularly in noisy cases. SimCO is used for dictionary learning. This SimCO method simultaneously updates an arbitrary number of atoms. The problem of increased singular vectors in K-SVD can be minimized by using a regularized version of SimCO.

In this paper we have proposed adaptive patch-based dictionary learning by introducing a hybrid algorithm of SimCO along with FOCUSS for MR images. The process of eliminating the aliasing and noise is performed in one step while data fidelity is enforced in the next step. In both noisy and noiseless cases, our technique provides superior reconstruction of images in empirical results. The performance is confirmed by the collection of various sampling trials and *k*-space under-sampling aspects. Fast convergence with accurate reconstruction at high under sampling rates is achieved by the proposed algorithm.

The rest of this paper is as follows: In Section 2, compress sensing on MR images and learning sparsifying transform is reviewed. Problem formulation for MRI reconstruction using adaptive patch-based dictionary and the suggested algorithm with its properties are also described. Section 3 provides the empirical performance of our algorithm with several examples, using a diversity of sampling schemes. The conclusion is drawn in Section 4.

## 2. Materials and Methods

### 2.1. Compressed Sensing of MR images

A sparse signal is one that has many zeros and few nonzero values. The aim of the sparse approximation is to synthesize a given signal or measurement vector as a linear combination of a small number of sparsifying transform vectors $\;,\;\psi_{i}$.

MR image acquisition can be modelled as an under-sampled measurement of MR image in *k*-space with the help of measurement matrix  Φu. Mathematically, MR image x∈ℂq is encoded to a measurement vector z∈ℂm as
(1)  z=Φux ,
where Φu∈ℂm×q is under-sampled Fourier encoding or measurement matrix. Whenever the number of unknowns is greater than the number of *k*-space samples q> m, it is called under-sampling.

Compressed sensing (CS) provides a promising way of reconstructing x from its undersampled measurements z provided x is sparse in some sparsifying transform domain, Ψ, also known as the dictionary. The recovery of x can be formulated as l0 minimization of sparsified image Ψx [33,34]:(2) minx|| Ψx||0   s.t.   Φux=z ,
Ψ∈ℂt×q is a sparsifying transform such as wavelet or DCT or any other learned dictionary.

The disadvantage of the model in (2) is that the sparse coding step is NP (Nondeterministic Polynomial-time) hard because the algorithm involves $l_{0}-$quasi norm. The non-convex formulation of (2) can be transformed to a convex problem by using $l_{1}-$norm [35], i.e.,
(3)$$\underset{x}{\;\min}{\left||\Phi_{u}x-z|\right|}_{2}^{2}+\gamma \;{\left||\Psi x|\right|}_{1}$$
where γ is the Lagrangian multiplier.

CS has been successfully applied in the recovery of static and dynamic MR images [12,13,15,16] but in this article we focus on static MR images and study them in detail.

### 2.2. Learning Sparsifying Transform

The quality of the reconstructed image mainly relies on the sparsifying transform. The main constraint of high undersampling in nonadaptive compressed sensing of MR images is solved by adaptive dictionary updates with high sparsity. In our framework, we use patch-based adaptive dictionary learning. For this purpose, let the given image x∈ℂq be denoted as a combination of patch vectors xij∈ℂn of 2D squared image where dimension of each patch is  (n ×n) pixels.  i,j marks the position of a patch starting from top left corner of the image. D∈ℂn×K  denotes the patch-based dictionary having K number of atoms and θij∈ℂK is the sparse representation of xij patch with respect to D. Dictionary D is said to be over complete when n<K.

The following optimization problem solves the patch-based dictionary learning as
(4) minD,G ∑ij||Wijx− Dθij||22   s.t.||θij||0≤τ0,   ∀i,j. 
where  G represents the set {θij }ij of sparse approximation of all patches and τ0 is the required sparsity. Wij∈ℝn×q matrix acts as an operator that brings out the patch xij from given image x such as
(5)xij=Wijx,
This learning scheme in (4) helps to minimize the total fitting error of all image patches while learning the dictionary, subject to sparsity restrictions.

The optimization formulation used in (4) is NP hard for the fixed D and can be solved with many algorithms like MOD, K-SVD or SimCO [21,22,26]. Such types of algorithms normally alternate between finding the dictionary D, and the sparse representations G. Most researchers use the K-SVD algorithm, in which the dictionary atoms are updated sequentially along with related sparse coefficients for the patches. Since singular value decomposition (SVD) is used *k*-times in this algorithm for updating the k-atoms of the dictionary sequentially, hence called K-SVD.

### 2.3. Problem Formulation

Reconstructed compressively sampled biomedical images typically suffer from numerous artifacts on high under sampling factors. Undersampling of *k*-space and noise in samples are two main causes of artifacts. A decent dictionary must be capable of minimizing the artifacts which are noticed in zero filled Fourier reconstruction and be consistent to produce reconstructed images using available *k*-space data. Possible cost function is as follows:(6)$$\underset{D,G,x}{\;\min}\;\underset{ij}{\sum }\;{\left||x_{ij}-\;D\theta_{ij}|\right|}_{2}^{2}+\;\eta \;{\left||\Phi_{u}x-z|\right|}_{2}^{2}\;s.t.\;{\left||\theta_{ij}|\right|}_{0}\leq \tau_{0},\;\forall i,j$$
In (6), the 1st term is responsible for the quality of the sparse approximation of the patched images with respect to the dictionary D whereas the 2nd term enforces data consistency in *k*-space. Parameter  η depends on standard deviation σ of measurement noise such as $\eta =\left(\;\frac{\lambda }{\sigma }\;\right)$ and λ is taken as a positive constant.

Our cost function is capable of learning an adaptive dictionary to reconstruct the underlying image but it is NP hard and non-convex even when $l_{0}-$ quasi norm is relaxed to $l_{1}-$ norm. Hence we use alternate minimization methods to solve this problem. Elimination of aliasing and noise is done through the adaptive patched sparsity, whereas the elimination of the artifact is done from overlapping patches.

### 2.4. The SiFo Algorithm

Adaptive learning of sparsifying transforms is a highly non-convex problem and is computationally expensive. Typically the problem in (6) is solved in two steps. (i) Dictionary learning and sparse coding are updated alternately keeping the estimated signal x fixed while in second step (ii) the estimated signal x is updated to satisfy the data fidelity while keeping the dictionary and sparse representation fixed.

#### 2.4.1. Updating the Dictionary and Sparse Coding

Since x is fixed, the objective function in (6) becomes
(7)$$\underset{D,G}{\;\min}\;\underset{ij}{\sum }{\left||x_{ij}-\;D\theta_{ij}|\right|}_{2}^{2}\;s.t.\;{\left||d_{k}|\right|}_{2\;}=1\;\forall k\;\&\;{\left||\theta_{ij}|\right|}_{0}\leq \tau_{0},\;\forall i,j$$
Extra constraint of unity norm on dictionary atoms ($d_{k}$,1 ≤k≤K) is applied to avoid scaling issues [36]. Many researchers use the K-SVD to learn the dictionary where the atoms of the dictionary are updated one by one i.e., K-times SVD which increases the computation time. Saiprasad et al. [22] performed tremendous work on MR image reconstruction from highly under-sampled *k*-space data using the K-SVD technique to learn the dictionary and a greedy algorithm such as orthogonal matching pursuit (OMP) for updating sparse coefficients. His work in DLMRI showed noticeable improvements in the reconstruction of different medical images along with other performance parameters such as SNR and high frequency error numbers (HFEN). He compared his results with Lusting et al [12] (denoted as LDP). One main problem in OMP is that it imposes the hard constraint [28] to achieve a sparse solution. This hard sparseness limitation may not be suitable for medical imaging applications because fast and abrupt changes of the image values can occur depending on the support setting up visible and irritating high frequency artifacts. Another problem for the OMP algorithm is that it is not very computationally efficient.

We have used the focal underdetermined system solver (FOCUSS) for updating the sparse representation matrix. Since the FOCUSS introduces the sparseness of the image as a soft constraint, high frequency artifacts are also minimized as compared to OMP because the non-zero image values are progressively suppressed. FOCUSS also inclines to suppress the reconstruction noise because aliasing artifacts and noises are usually isolated; thus, these artifacts and noises can be easily removed during the pruning process of FOCUSS. The empirical results (in Section 3) show better outcomes, especially in a noisy case. Lastly, FOCUSS can be applied computationally in an efficient manner by means of successive quadratic optimization. This is a relatively significant advantage over computationally expensive sparse optimization algorithms such as OMP.

In our framework, we use SimCO [26] to learn the dictionary D. The key characteristic of SimCO is to update all the atoms and corresponding non-zero coefficients simultaneously and hence reduce the computation cost.

Because of the above mentioned advantages of SimCO and FOCUSS for the reconstruction of MR images, we termed our framework “SimCO plus FOCUSS” the SiFo. Experimental results on synthetic as well as in vivo data determine improved performance of our algorithm even from highly sparse *k*-space samples and show better results than Saiprasad et al [22].

##### SimCO

SimCO is considered to train a dictionary having unit norm atoms from a set of signals so that signals can be better represented as the linear combination of a few atoms of the dictionary.

The optimization problem to update the dictionary for regularized SimCO [26] is as follows
(8)$$\underset{D,G}{\min}\;\underset{ij}{\sum }{\left||x_{ij}-\;D\theta_{ij}|\right|}_{2}^{2}+µ\;{\left||\theta_{ij}|\right|}_{2}^{2},$$
where µ (µ>0) is a properly chosen constant for regularized SimCO, D is the dictionary learned on input patches for sparse representation $\;\theta_{ij}$. To solve the optimization problem (8), SimCO follows the two-steps optimization process, involving sparse coding and a dictionary update. The sparse coding step involves estimating the sparse representations $\theta_{ij}$ of the signal patches $x_{ij}$ for the given dictionary D.

In our proposed SiFo framework, we are using FOCUSS as a sparse coding step for SimCO. In the dictionary update step, SimCO uses the optimization methods on manifolds to learn the dictionary D and $\theta_{ij}$ simultaneously, while keeping sparsity pattern of $\theta_{ij}$ unchanged. So this framework is capable of updating the multiple atoms of D simultaneously in each iteration and guarantees the atoms of D to have the unit $\;l_{2}-$norm.

##### FOCUSS

FOCUSS is a sparse coding technique based on minimum norm optimization that iteratively finds the sparse solution using the weighting matrix M. To find sparse vector $\theta_{ij}$, we formulate it as
(9)$$\theta_{ij}\;=Mb\;,\;\forall i,j$$
here M is a weighting matrix, and b is calculated as follows:(10)$$\;\min{\left||\;b|\right|}_{2}^{2}\;s.t.\;DMb=x_{ij},\;\forall i,j\;$$
The optimal solution is:(11)$$b={\left(DM\right)}^{†}\;x_{ij},\;\forall i,j$$
(12)$$\theta_{ij}=Mb=M{\left(DM\right)}^{†}x_{ij},\;\forall i,j$$
The uniqueness of FOCUSS is that the weighting matrix M is continuously updated. In FOCUSS, pruning the process of sparse representation is very important to guarantee the performance of the algorithm. FOCUSS starts by finding an initial estimate of the sparse signal to initialize the weighting matrix M at the starting of the iteration and then this solution is pruned to a sparse signal representation iteratively. FOCUSS removes the noise during reconstruction due to this pruning process. Let l be the current iteration of the algorithm, then the basic FOCUSS algorithm is composed of the following 3 steps:(13)$$\mathrm{Step}1:\;\;\;M_{l}=\mathrm{diag}\left({\left|{\left(\theta_{ij}\right)}_{l-1;1}\right|}^{p},{\left|{\left(\theta_{ij}\right)}_{l-1;2}\right|}^{p},\ldots ,\;{\left|{\left(\theta_{ij}\right)}_{l-1;k}\right|}^{p}\right),\;\forall i,j$$
(14)$$\mathrm{Step}2:\;b_{l}={\left(DM_{l}\right)}^{\forall }x_{ij},\;\forall i,j$$
(15)$$\mathrm{Step}3:\;{\left(\theta_{ij}\right)}_{l}=M_{l}b_{l},\;\;\;\forall i,j$$
After computing ${\left(\theta_{ij}\right)}_{l}$ in Step 3, the weighting matrix is re-computed, and FOCUSS iteration, Steps 1 to 3 is reapplied [28]. The parameter p is given by $\left(0.5<p<1\right).$

#### 2.4.2. Updating the Estimated Image(s) for Reconstruction

To update the reconstruction image x, keep the dictionary and the sparse representation constant then the sub-problem for our cost function in (6) can also be written as follows:(16)$$\underset{x}{\min}\;\underset{ij}{\sum }{\left||W_{ij}x-\;D\theta_{ij}|\right|}_{2}^{2}+\eta \;{\left||\Phi_{u}x-z|\right|}_{2}^{2}$$
The formulation in (16) is the least square problem and detail solution is in Appendix A. The solution is as follows [22].
(17)$$\Phi x\left(k_{x},k_{y}\right)=\left\{\begin{array}{l} \begin{array}{c} N\left(k_{x},k_{y}\right)\;\;\;\;\;\;\;\;\;\;\;\;\;\;\;\;\;\;\;\;\;\;\;\;\;\;\;\;\;\;\;\;,\left(k_{x},k_{y}\right)\;\notin ℧\\ \frac{N\left(k_{x},k_{y}\right)+\eta N_{0}\left(k_{x},k_{y}\right)}{1+\eta }\;,\left(k_{x},k_{y}\right)\in ℧ \end{array} \end{array}\right.,$$
Here x, is reconstructed by taking the IFFT of  Φx.

From Equation (A9) in Appendix A
(18)$$N=\Phi \underset{ij}{\sum }\begin{array}{c} W_{ij}^{T} \end{array}\;D\;\theta_{ij}\frac{1}{\alpha }\;,$$
(18) is called the “patched average result” in Fourier domain and $\Phi x\left(k_{x},k_{y}\right)\;$ represents the updated value on location $\;\left(k_{x},k_{y}\right)$ of the *k*-space. $N_{0}=\Phi \;\;\Phi_{u}^{H}z\;$ represents zero filled *k*-space measurement and ℧ denotes the subset of *k*-space that has been sampled.

Process of reconstruction MR images, from undersampled *k*-space measurements using adaptive dictionary, is described in Algorithm 1.


**Algorithm 1:**
**Goal:** To learn the dictionary for reconstruction of under-sampled image**Input:**
z = training signal in *k*-space measurements, μ, p**Output:**
x^= An estimated reconstruction MR Image**Initialization:**
x = x0 =   ΦuHz Main Iteration:
Alternately learn dictionary by SimCO and sparse (coding) approximation for x patches by FOCUSSUpdate xN←  FFΤxRestore sampled frequency to update the N as per (17)x^← ΙFFΤN

A more extensive pseudocode is presented in Appendix B.

## 3. Results and Discussion

One of the important factors in dictionary learning is its initialization. In these experiments, we initialize our dictionary from a subset of image patches.

In our experiment we take the image in a Fourier domain, and apply our proposed framework for noiseless and noisy scenarios. The image in Fourier data is processed with different sampling schemes. These images are converted to over lapping patches of size $\;\left(\sqrt{n}\times \sqrt{n}\right)\;$, in our case  n=36. From these patches, we have initialized the data for dictionary learning.

We update the dictionary through regularized SimCO and sparse coefficients using FOCUSS. Through different iterations, our proposed algorithm reconstructs the image properly and in doing so outperforms the DLMRI proposed by Saiprasad et al. The DLMRI technique based on KSVD has outperformed several methods such as MOD and LDP (method by Lusting et al). Therefore in our comparison, we compared our method with only DLMRI. Since our proposed method performs better than the DLMRI, hence it will also perform better than any other CSMRI technique. So our method SiFo for reconstruction is compared with a leading DLMRI method by Saiprasad et al.

The performance of the proposed algorithm is validated with various under sampling factors, for noiseless and noisy cases. The undersampling is directly applied on *k*-space (fully-sampled) MR data set. Axial T2-weighted reference images of the brain are used as MR images, taken from vivo MR scans of size 512×512 from American Radiology Services.

All implementations were coded in Matlab 9.2.0.538062 (R2017a). The Computations were performed with 7th Generation Intel Core i5-7200U Processor (2 Cores-4 Threads).

During the tests with noisy and noiseless images, we fixed values of different parameters such as atoms  K=n= 36, sparsity $\tau_{0}$ = 6, γ = 140, maximum overlapped patch called overlap stride “*r*” as r=1 (the distance in pixels between the corresponding pixel position in adjacent image patches.), regularized parameter for SimCO is μ = 0.05 and for FOCUSS diversity measure  p=0.5. Both SiFo and DLMRI are performed for 15 iterations with fixed sparsity $\tau_{0\;}$ and 200xK patches.

The SiFo and DLMRI learning techniques need an initialization as discussed above for the dictionary [20]. Real valued sparsifying transforms [37] were used in the simulated experiments with real valued images. The reconstruction quality is computed through PSNR and with the high frequency error norm (HFEN). PSNR is calculated, normally defined in decibels (dB), as the ratio of the maximum possible intensity level of the original image to the root mean square (RMS) compressed/reconstruction error relative to the original image. On image compression, it is considered as a standard image quality measure and is being used in compressed sensing MRI beforehand [38], along with the associated metric of the signal to noise ratio (SNR in dB) [14,39]. Reconstruction of edges quality and fine features are measured by (HFEN) which is calculated as the norm of the result acquired by Laplacian of Gaussian (LoG) filtering, the difference between the reconstructed and reference images.

Additionally we have also considered some quantitative measurements for image comparison like a correlation, similarity index (SSIM: Structural SIMilarity) and sharpness with reference to the original image in all noisy and noiseless cases. High fidelity can be measured with direct comparisons between the original image and reconstruction with image subtraction. All performance tests are employed on Shepp-Logan Phantom as well as real brain MRI data.

### 3.1. Performance in Noiseless Scenario

We first compared our proposed method with the DLMRI in a noiseless case. Figure 1 shows the performance of algorithm on brain and phantom image in noiseless scenario. Algorithm performance on a phantom and a brain image is evaluated using a 2D variable density random sampling of *k*-space. The dictionary learning algorithm (SiFo) reconstructed both the images free from artifacts and the aliasing effect. The results were achieved by running our algorithm for 15 iterations. The reconstruction with SiFo algorithm is clearer and sharper than with DLMRI.

#### 3.1.1. PSNR

From the Figure 1 and Table 1, the comparison of PSNR for both methods can be analyzed which shows that SiFo converges quickly by using $l_{2}-$norm reconstruction of difference between two successive iterations. So our algorithm is appropriately minimizing the noise and aliasing observed in the zero filled case to deliver a better reconstruction.

#### 3.1.2. HFEN

The SiFo performs better to capture the image of brain and phantom with fast convergence than DLMRI. This can be easily observed from Figure 1 and Figure 2, where our method showed better edges and sufficient features of reconstruction images.

#### 3.1.3. Correlation/Similarity Index/Sharpness

Although the correlation and similarity index display marginal improvement in the reconstruction of images for a brain and phantom in a noiseless case, the sharpness indicates good results as shown in Table 1.

### 3.2. Performance in Noisy Scenario

In this case, we added zero-mean white Gaussian fixed noise of standard deviation for all cases such that sigma = 10.2489 in *k*-space data. In the course of reconstruction update stage of the algorithm in (17), the noisy scenario involves weighted averaging in *k*-space. The performance of our technique on the fully sampled noisy image is observed by using a different sampling mask. Our process of reconstruction has sufficiently removed the noise and aliasing observed in the zero-filled result.

#### 3.2.1. PSNR

From Figure 3, the comparison of PSNR for both methods is observed for the reconstruction of the MR image. Our algorithm significantly removes aliasing and noise noticed in the zero filled result, hence providing a better reconstruction. The reconstruction error magnitude of the image for SiFo displays pixel errors of considerably reduced magnitude and fewer structure than that of DLMRI technique by Saiprasad et al. [22].

Similarly, the reconstruction of the phantom image [40] has shown the same improved result as the reconstruction of the brain MR image.

#### 3.2.2. HFEN

The convergence rate has been observed to be efficient. If we compare the SiFo with DLMRI, the SiFo converges at the rate 0.95 whereas DLMRI stops at 1.14 (in the reconstruction of brain image) at the end of the executed number of iteration 15. This shows that the proposed algorithm outperforms DLMRI regarding reasonable noise. So image features for the reconstruction of the brain and phantom images are smooth, clear and free from the effect of aliasing and artifacts as shown in Figure 4.

#### 3.2.3. Correlation/Similarity index/Sharpness

Although the correlation and similarity index show a slight improvement in the reconstruction of the brain and phantom images in the noisy case, sharpness indicates noticeable improvement as shown in Table 2 and Table 3. Brightness or sharpness of an MRI scan depends on the relaxation time of the specific molecules.

Graphs in Figure 5a,b are shown for a noisy case with 4 fold Cartesian under sampling for reconstruction of a phantom image.

Performance comparison of SiFo vs DLMRI for noisy brain images for radial sampling are shown in Figure 6 and Table 3.

## 4. Conclusions

In this paper, adaptive patch-based dictionary learning framework is proposed by introducing a hybrid algorithm of SimCO and FOCUSS for MR images. The algorithm is based on optimization of manifold and permits a simultaneous update of all atoms and corresponding coefficients. Artifacts such as the aliasing and noise are removed in one step, while enforcing the data fidelity in the next step. Our technique has shown the superior reconstruction of images in the empirical results for noisy and noiseless cases. The performance is validated by using different sampling masks and *k*-space under sampling ratios. Fast convergence with more accurate reconstruction at high undersampling is achieved by this scheme with robust performance in a noisy environment. However, the convergence in FOCUSS still needs to be improved. The proposed framework may be implemented on other medical imaging problems which will be considered for future research. A promising prospect involves exploring the application of the proposed methodology for other types of noise or artifacts that come during scanning time like head movement, etc.

## Figures and Tables

**Figure 1 sensors-19-01918-f001:**
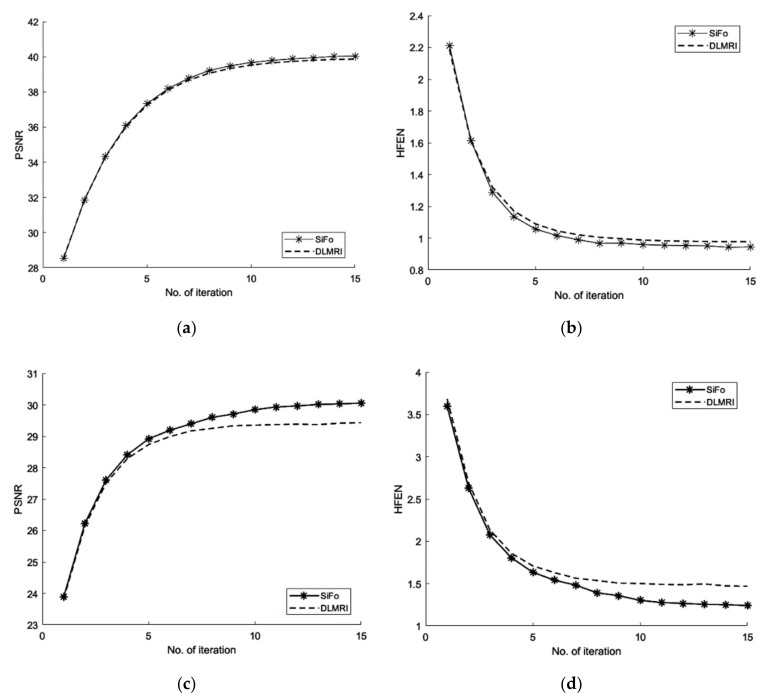
Algorithm performance in a noiseless case. (**a**) PSNR vs. iterations with comparison to DLMRI for a brain image; (**b**) HFEN vs. iterations with comparison to DLMRI for a brain image; (**c**) PSNR vs. iterations with comparison to DLMRI for a phantom image; (**d**) HFEN vs. iterations with comparison to DLMRI for a phantom image.

**Figure 2 sensors-19-01918-f002:**
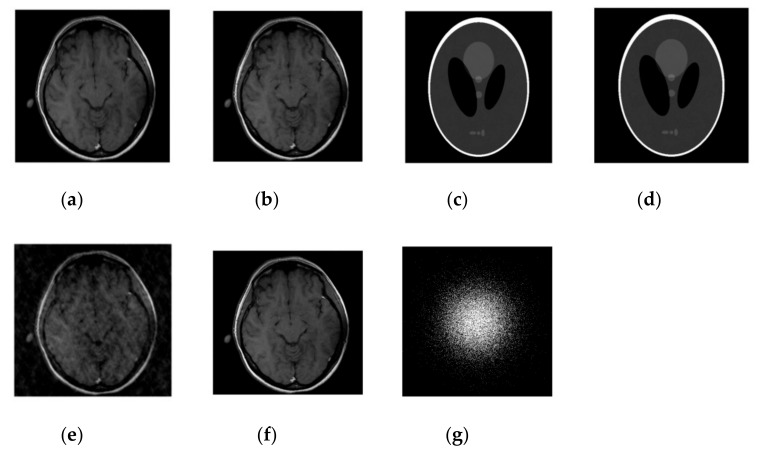
Images recovery for noiseless case. (**a**) Recovered MR image of brain by SiFo; (**b**) Recovered MR image of brain by DLMRI; (**c**) Recovered MR image of phantom by SiFo; (**d**) Recovered MR image of phantom by DLMRI; (**e**) Reconstruction brain image with zero filling; (**f**) Reference MR image for brain; (**g**) *k*-space sampling mask with 10 fold.

**Figure 3 sensors-19-01918-f003:**
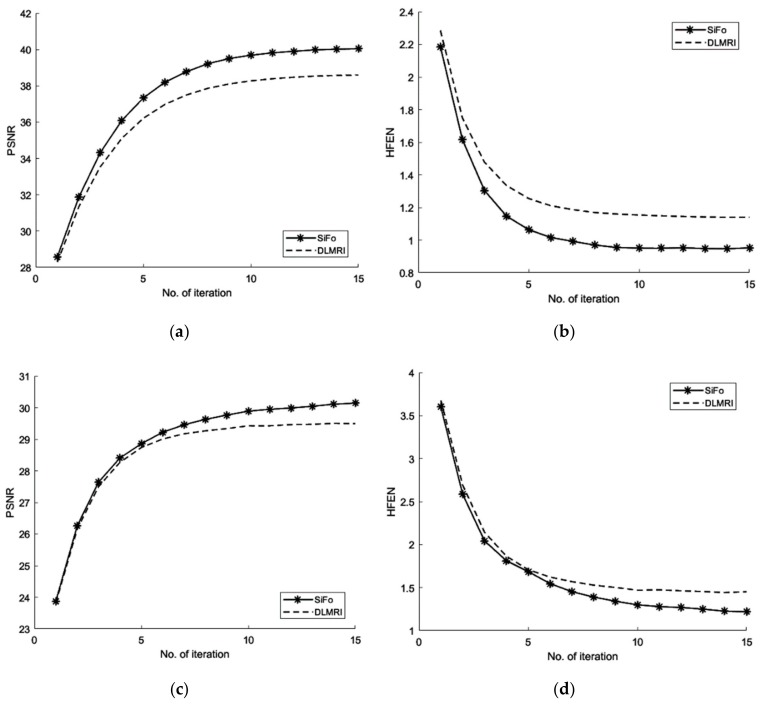
Algorithms performance for a noisy case. (**a**) PSNR vs iterations with comparison to DLMRI for brain image; (**b**) HFEN vs iterations with comparison to DLMRI for a brain image; (**c**) PSNR vs iterations with comparison to DLMRI for a phantom image; (**d**) HFEN vs iterations with comparison to DLMRI for a phantom image.

**Figure 4 sensors-19-01918-f004:**
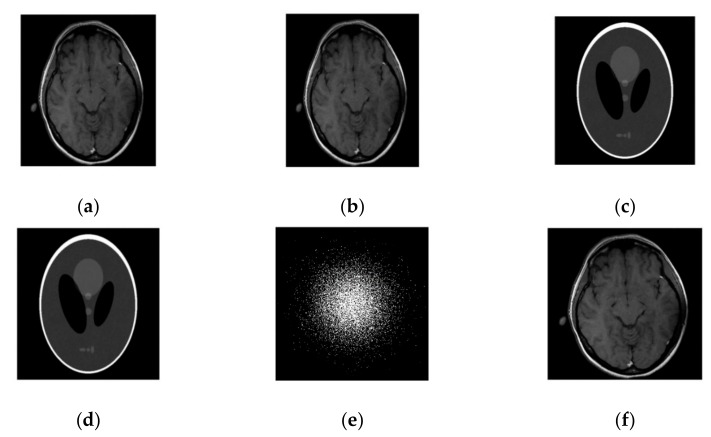
Images recovery in a noisy case. (**a**) Recovered MR image of a brain by SiFo; (**b**) Recovered MR image of brain by DLMRI; (**c**) Recovered MR image of phantom by SiFo; (**d**) Recovered MR image of a phantom by DLMRI; (**e**) sampling mask in *k*-space with 10 fold; (**f**) Reference MR image of a brain.

**Figure 5 sensors-19-01918-f005:**
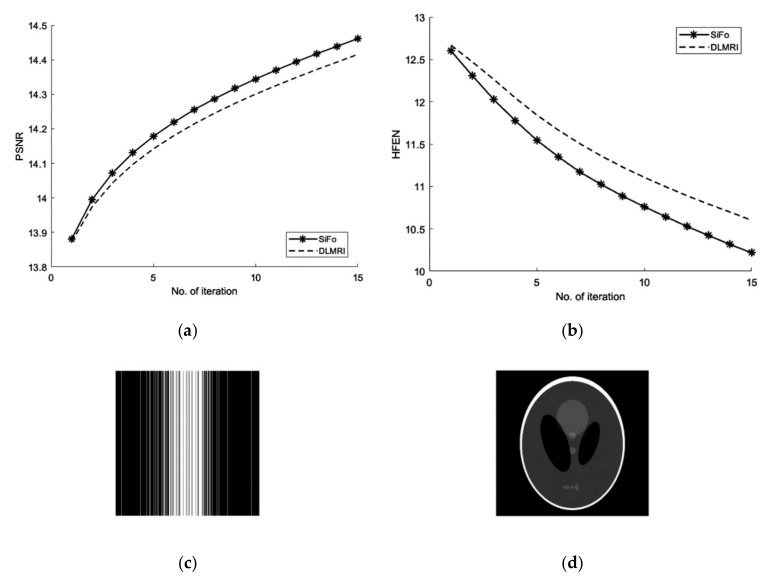
Algorithms performance in noisy case with cartesian sampling. (**a**) PSNR vs iterations with comparison to DLMRI for a phantom image (**b**) HFEN vs iterations with comparison to DLMRI for a phantom image (**c**) Cartesian sampling scheme with 4 fold. (**d**) Recovered image.

**Figure 6 sensors-19-01918-f006:**
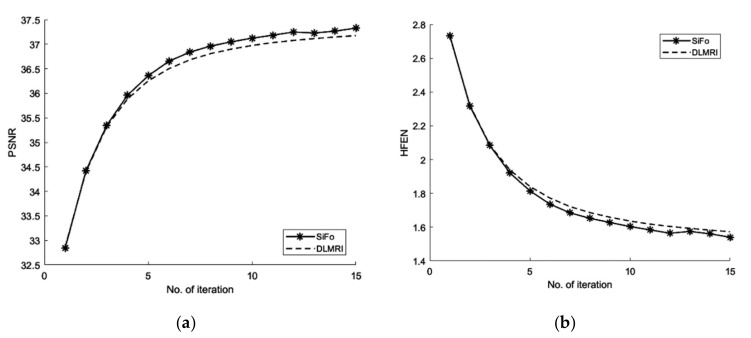
Algorithms performance of SiFo vs DLMRI in a noisy case with radial sampling mask. (**a**) PSNR vs iterations for a phantom image; (**b**) HFEN vs iterations for a phantom image; (**c**) Recovered MR image of a brain by SiFo; (**d**) Recovered MR image of a brain by DLMRI; (**e**) Radial sampling mask in *k*-space with a 6.1 fold undersampling.

**Table sensors-19-01918-t001a:** (**a**)

Parameters	DLMRI	SiFo	Difference	Improvement (%)
Correlation	0.998	0.9981	0.00010	0.01
Similarity Index (SSIM)	0.8899	0.8935	0.0036	0.04
Sharpness	944.1884	984.3163	40.1279	4.25

**Table sensors-19-01918-t001b:** (**b**)

Parameters	DLMRI	SiFo	Difference	Improvement (%)
Correlation	0.988	0.9896	0.0016	0.16
Similarity Index (SSIM)	0.8151	0.8492	0.0341	4.18
Sharpness	3739	4434.4	695.4	18.6

**Table sensors-19-01918-t002a:** (**a**)

Parameters	DLMRI	SiFo	Difference	Improvement (%)
Correlation	0.9975	0.9981	0.0006	0.06
Similarity Index (SSIM)	0.8282	0.8902	0.062	7.49
Sharpness	870.9357	980.2718	109.3361	12.5

**Table sensors-19-01918-t002b:** (**b**)

Parameters	DLMRI	SiFo	Difference	Improvement (%)
Correlation	0.9876	0.9892	0.0016	0.16
Similarity Index (SSIM)	0.7514	0.7716	0.0202	2.69
Sharpness	870.9357	4177.1	676.9	19.3

**Table 3 sensors-19-01918-t003:** Performance parameter of Algorithm with noisy case for Brain Image with Radial sampling.

Parameters	DLMRI	SiFo	Difference	Improvement (%)
Correlation	0.9965	0.9966	0.00010	0.01
Similarity Index (SSIM)	0.7974	0.8012	0.0038	0.48
Sharpness	688.437	720.039	31.6017	4.60

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
