# Peer review of "Improved Reconstruction of MR Scanned Images by Using a Dictionary Learning Scheme"

_sensors, 2019, doi:10.3390/s19081918_

Round 1
Reviewer 1 Report
Presented is an analysis which explore, in short, a method that could permit for greater fidelity of MR images from k-space using a proposed dictionary learning scheme method of compressive sensing. Other methods are shown as a comparison and demonstrative samples shown with and without noise for a Shepp-Logan and sample brain MR image. This analysis is important and has merit, not just as an alternative outperforming method of signal reconstruction but as one that, practically, could mean a usable radiologically valuable image is obtained in less time or with less gainful equipment. For practical users of MR and the manufacturers who employ standard methods in reconstruction, this method could be another one employed assuming the quality of the image is appropriate for its clinical use.
That being said, there are several methods the authors should address and, if they chose not to perform, why. In discussing how their SiFo method performs on noise, more detail should focus on the nature of noise germane to MRI imaging (especially in frequency domain). The most common types of noise or artifact from a scanner would come from head movement. I would encourage the authors to investigate and comment on this, as one of the precluding factors of this method may be how SiFo's method deals with that appreciable metric of noise.
Additionally, is there any runtime performance comparison for this method? Even if marginal, the authors should report this as, if this method were to be practically applied but takes appreciably longer than existing methods, it could prohibit meaningful application of this in a clinical setting (also, it's just good to have).
As for the figures and images, please include the type of MRI image used (I suspect a T1-weighted MPRAGE or FAST or SPGR or something like that? information about sampling, acquisition parameters, other factors like relaxation time flip angle etc...) These images need to be be bigger to be meaningful. I respect this is a manuscript however I am straining to tell differences. Additionally, please include relative difference in the tables for the correlation, similarity, ssim and sharpness.
More extensive pseudocode in the appendix for the algorithm would be even more helpful, but I defer to the authors whether this is necessary.
This is important method to explore which was done adequately, and with some work could be a meaningful contribution to MR from both the signals and acquisition standpoints.
Author Response
Response to Reviewer 1 Comments
Presented is an analysis which explores, in short, a method that could permit for greater fidelity of MR images from k-space using a proposed dictionary learning scheme method of compressive sensing. Other methods are shown as a comparison and demonstrative samples shown with and without noise for a Shepp-Logan and sample brain MR image. This analysis is important and has merit, not just as an alternative outperforming method of signal reconstruction but as one that, practically, could mean a usable radiologically valuable image is obtained in less time or with less gainful equipment. For practical users of MR and the manufacturers who employ standard methods in reconstruction, this method could be another one employed assuming the quality of the image is appropriate for its clinical use.
Response: Many thanks for the encouraging and valuable comments on our submitted manuscript.
Reviewer 1 Point 1:
That being said, there are several methods the authors should address and, if they chose not to perform, why. In discussing how their SiFo method performs on noise, more detail should focus on the nature of noise germane to MRI imaging (especially in frequency domain). The most common types of noise or artifact from a scanner would come from head movement. I would encourage the authors to investigate and comment on this, as one of the precluding factors of this method may be how SiFo's method deals with that appreciable metric of noise.
Response: Agreed. The suggested relevant literature has been incorporated in the revised manuscript to enhance the motivation of the study and consequently it helps a lot to improve the technical strength of the manuscript.
· DLMRI techniques, based on KSVD, had outperformed several methods like MOD, zero-filling and LDP (Lusting et al). Other CSMRI methods reviewed in introduction offer only small improvement over DLMRI and hence not included in in our comparison. So our proposed method is compared with a leading DLMRI method (by Saiprasad Ravi Shankar et al) which had already outperformed above mentioned techniques.
· In our experiment, raw MR data is already stored in k-space static image .Therefore in this paper; we have restricted our attention to compressed sensing (CS) for static MRI and study it in detail. Since undersampling artifacts are addressed in this research work so motion artifacts (like head movement noise) during acquiring data from the MRI scanner are beyond the scope of this paper. Please see the section (2.1. Compressed Sensing of MR images) of the revised manuscript. However it is very useful suggestion and we have incorporated this aspect in the future work of the revised manuscript. (Please see the 4.conclusion).
· Zero-mean white Gaussian noise was added in k-Space data as done by Saiprasad et al in DLMRI technique (Please see section 3.2. Performance with the Noisy Scenario of the revised manuscript).
· FOCUSS has a tendency to suppress the reconstruction noise because the aliasing artifacts and noises are usually isolated; therefore, these noises can be easily removed during the pruning process of FOCUSS. Please see the section (2.4.1. (b). FOCUSS) of the revised manuscript.
Reviewer 1 Point 2:
Additionally, is there any runtime performance comparison for this method? Even if marginal, the authors should report this as, if this method were to be practically applied but takes appreciably longer than existing methods, it could prohibit meaningful application of this in a clinical setting (also, it's just good to have).
Response: Taking it as a proof of concept, the proposed method is for “off line reconstruction for MR image” tested with images having size (512 x 512). The proposed method takes 40 sec more than that of DLMRI per iteration. However the recovery time could be optimized further and may leave for future work.
Reviewer 1 Point 3:
As for the figures and images, please include the type of MRI image used (I suspect a T1-weighted MPRAGE or FAST or SPGR or something like that? information about sampling, acquisition parameters, other factors like relaxation time flip angle etc...) These images need to be bigger to be meaningful. I respect this is a manuscript however I am straining to tell differences. Additionally, please include relative difference in the tables for the correlation, similarity, ssim and sharpness.
Response: Agreed. Axial T2-weighted reference image of the brain is used in our experiments. Please see the section (3. Results and Discussion) of the revised manuscript.
Sampling mask information in k-space for undersampling scheme is mentioned in the figures (2g, 4e, 5c and 6e) in the revised manuscript.
Additionally, the relative percentage difference in the Tables (1-3) for the correlation, similarity Index (SSIM) and sharpness has been incorporated in the revised manuscript as suggested.
Reviewer 1 Point 4:
More extensive pseudocode in the appendix for the algorithm would be even more helpful, but I defer to the authors whether this is necessary
Response: Agreed: We respect your opinion in this regards and we have incorporated the suggested pseudocode in the appendix-B for the algorithm in the revised manuscript.
This is important method to explore which was done adequately, and with some work could be a meaningful contribution to MR from both the signals and acquisition standpoints.
Response: Thank you for valuable comments. Authors tried their level best to address all the valuable comments of respected reviewer in the revised manuscript so that it would be a more meaning full contributions.
Note: All the necessary and concerned comments have been incorporated in the revised manuscript.
----------------------------------------------------------------------------------------------------------------
Author’s Response: Many thanks to anonymous reviewer for his valuable comments, suggestions and time on our submission. These comments and suggestions help us considerably to improve the quality of revised manuscript in term of technical contributions and presentation
Reviewer 2 Report
The authors described a method of MRI reconstruction using dictionary learning. There are some merits of the study however, the work may need some improvement:
1. The authors ignored the latest deep learning based MRI reconstruction. The following papers can be good references:
Yang, Guang, et al. "DAGAN: deep de-aliasing generative adversarial networks for fast compressed sensing MRI reconstruction." IEEE transactions on medical imaging 37.6 (2018): 1310-1321.
Seitzer, Maximilian, et al. "Adversarial and perceptual refinement for compressed sensing MRI reconstruction." International Conference on Medical Image Computing and Computer-Assisted Intervention. Springer, pp. 232-240, Cham, 2018.
Schlemper, Jo, et al. "Stochastic Deep Compressive Sensing for the Reconstruction of Diffusion Tensor Cardiac MRI." International Conference on Medical Image Computing and Computer-Assisted Intervention. Springer, pp. 295-303, Cham, 2018.
2. The proposed method should be compared with at least one of the deep learning based method, e.g., the Cascaded CNN.
3. 2D undersampling used in the study is not realistic. Normally only the phase-encoding direction gain the acceleration.
4. Are the results of the compared method got statistical significance?
Author Response
Response to Reviewer 2 Comments
Reviewer 2 Point 1:
The authors ignored the latest deep learning based MRI reconstruction. The following papers can be good references:
Yang, Guang, et al. "DAGAN: deep de-aliasing generative adversarial networks for fast compressed sensing MRI reconstruction." IEEE transactions on medical imaging 37.6 (2018): 1310-1321.
Seitzer, Maximilian, et al. "Adversarial and perceptual refinement for compressed sensing MRI reconstruction." International Conference on Medical Image Computing and Computer-Assisted Intervention. Springer, pp. 232-240, Cham, 2018.
Schlemper, Jo, et al. "Stochastic Deep Compressive Sensing for the Reconstruction of Diffusion Tensor Cardiac MRI." International Conference on Medical Image Computing and Computer-Assisted Intervention. Springer, pp. 295-303, Cham, 2018.
Response: Agreed. The suggested recent literature references on deep learning based MRI reconstruction along with necessary remarks have been added in the introduction part of the manuscript. Consequently, the motivation of the study and technical strength of the manuscript have enhanced with this revision.
Please see the section (1.Introduction) of the revised manuscript.
Reviewer 2 Point 2:
The proposed method should be compared with at least one of the deep learning based method, e.g., the Cascaded CNN.
Response: Although, latest deep learning based MR image reconstruction has outperformed the conventional CS-MRI and DLMRI technique. But the main bottle neck is the availability of high computation resources and large amount of data which are not available in all cases. In those scenarios, dictionary learning based methods are beneficial since they outperform other conventional MRI reconstruction method trained on small sets of image data. We have also performed one experiment part, where CNN or cascaded CNN trained on one image/slice, such as in our proposed case, which has worst performance as compared to CS method. As we have worked on limited data, therefore proposed technique is compared with DLMRI.
Reviewer 2 Point 3:
2D undersampling used in the study is not realistic. Normally only the phase-encoding direction gain the acceleration
Response: Agreed. Normally centre dense (mask) under-sampling is used for simulation purpose and this mask undersamples the image data randomly which is not realistic. We have shown this in figures (2g, 4e). This under-sampling scheme is vastly used in CSMRI literature (Lusting, Saiprasad, Candes etc. et al) to validate their proposed methodology. We have also used 2D undersampling mask (centred dense) to validate our proposed method. We want to refer the figures (5c and 6e) to our worthy reviewer where Cartesian and radial undersampling is used which possess both frequency and phase encoding. Cartesian and radial trajectories are used routinely in clinical practice in MR imaging and their 2D undersampling may be considered realistic.
Reviewer 2 Point 4:
Are the results of the compared method got statistical significance?
Response: Yes. Images recovered by both the methods (DLMRI and SiFo) were presented randomly to different experienced radiologists. They declared that the images recovered by our proposed method are better as compared to DLMRI. Since it is difficult for us to show visually the improvement/comparison from the recovered images, we have compared through statistical data with relative difference in the tables for the correlation, similarity index (SSIM) and sharpness
Note: All the necessary and concerned comments have been incorporated in the revised manuscript.
----------------------------------------------------------------------------------------------------------------
Author’s Response: Many thanks to anonymous reviewer for his valuable comments, suggestions and time on our submission. These comments and suggestions help us considerably to improve the quality of revised manuscript in term of technical contributions and presentation.
Reviewer 3 Report
The authors propose a new reconstruction technique (termed SiFo) combining the SimCO technique (dictionary learning based denoising) with the so-called FOCUSS optimization algorithm. MR reconstruction/denoising experiments are presented on simulation (retrospective undersampling) for the noisy and noiseless cases. The proposed technique is compared to a deep learning MRI technique (DLMRI) in terms of SNR, SSIM and sharpness.
While the idea behind the technique is of interest to the community and clear efforts have been put in describing the technique, the English in the present manuscript is, unfortunately, of poor quality and requires major improvement. The poor English and grammar unfortunately affect the quality and reading of the paper and a major proofreading needs to be performed.
Moreover, I would invite the authors to consider the following points:
1) In this paper, simulations are performed by retrospectively undersampling a fully-sampled MR image. This step needs to be clarified. Do the authors apply the undersampling on a real acquired MR rawdata k-space (fully-sampled) or on a magnitude image (for example coming from Dicoms). While a simulation study is necessary, applying the proposed technique on real undersampled MR data would be of great interest. There are many groups sharing real dataset (see for example Dr. M. Lustig’s website, or Dr Tao Zhang’s page from Stanford, or many other researchers).
2) Please clarify if the added noise was Gaussian or Rician, in other words did you apply the noise on magnitude images or directly in the k-space?
3) Nowadays, most MR acquisitions are performed with multiple receiver coils (e.g., parallel imaging) and therefore applying the proposed technique in a parallel imaging fashion would approach more what is performed today in clinical routine and push up the quality of the paper.
4) Comparisons with previously published compressed sensing techniques are required. Many codes are available (e.g., Bart toolbox) and can easily be applied to the undersampled dataset.
In conclusion, while the proposed technique can be of interest for the MR community, the lack of good English is a major weakness of the paper, impacting its acceptance in its current form. Additional comparisons with previously published and validated CS techniques need to be performed and more realistic experiments (prospectively undersampled dataset, multi-coil dataset) need to be added.
Author Response
Response to Reviewer 3 Comments
Reviewer 3 Point 1:
In this paper, simulations are performed by retrospectively undersampling a fully-sampled MR image. This step needs to be clarified. Do the authors apply the undersampling on a real acquired MR rawdata k-space (fully-sampled) or on a magnitude image (for example coming from Dicoms). While a simulation study is necessary, applying the proposed technique on real undersampled MR data would be of great interest. There are many groups sharing real dataset (see for example Dr. M. Lustig’s website, or Dr Tao Zhang’s page from Stanford, or many other researchers).
Response: The undersampling is applied in k-space (fully-sampled) MR data set. We have worked on simulated data and focus on data set provided by Saiprasad Ravi Shankar (Data set reference provided by him: 2009, American Radiology Services [online].
http://www3.amercanradiology.com/pls/web1/wwimggal.vmg/
Please see the section (3, Results and Discussion) of the revised manuscript.
Reviewer 3 Point 2:
Please clarify if the added noise was Gaussian or Rician, in other words did you apply the noise on magnitude images or directly in the k-space?
Response: Zero-mean white Gaussian noise is added in k-Space data (see 3.2. Performance with the Noisy Scenario)
Reviewer 3 Point 3:
Nowadays, most MR acquisitions are performed with multiple receiver coils (e.g., parallel imaging) and therefore applying the proposed technique in a parallel imaging fashion would approach more what is performed today in clinical routine and push up the quality of the paper.
Response: At the moment we don’t have the access to the multi receiver coils data. However, we appreciate the feedback from respectable reviewer and we aim to test the proposed method on multiple receiver coil data as part of the future work.
Reviewer 3 Point 4:
Comparisons with previously published compressed sensing techniques are required. Many codes are available (e.g., Bart toolbox) and can easily be applied to the undersampled dataset.
Response: Dictionary Learning-based MRI (DLMRI) techniques, based on KSVD, had outperformed the several methods like MOD, zero-filling and LDP (method by Lusting et al). Other CSMRI method reviewed in introduction offer only small improvement over DLMRI and hence not included in in our comparison. So our proposed method is compared with a leading DLMRI method (by Saiprasad et al) which had already been shown to outperform the above mentioned techniques and our proposed technique has shown superior results than DLMRI.
In conclusion, while the proposed technique can be of interest for the MR community, the lack of good English is a major weakness of the paper, impacting its acceptance in its current form. Additional comparisons with previously published and validated CS techniques need to be performed and more realistic experiments (prospectively undersampled dataset, multi-coil dataset) need to be added.
Response: Thank you for the value able comment. The whole manuscript has been checked carefully to avoid any English/grammar mistakes.
Additionally, Dictionary Learning-based MRI (DLMRI) techniques, based on KSVD, had outperformed the several methods like MOD, zero-filling and LDP (method by Lusting et al). Other CSMRI method reviewed in introduction offer only small improvement over DLMRI and hence not included in our comparison. So our proposed method is compared with a leading DLMRI method (by Saiprasad et al) which has already been shown to outperform the above mentioned techniques and our proposed technique has shown superior results than DLMRI.
Note: All the necessary and concerned comments have been incorporated in the revised manuscript.
----------------------------------------------------------------------------------------------------------------
Author’s Response: Many thanks to anonymous reviewer for his valuable comments, suggestions and time on our submission. These comments and suggestions help us considerably to improve the quality of revised manuscript in term of technical contributions and presentation.
Round 2
Reviewer 1 Report
I encourage the authors to go-over the grammar and styling of the paper with someone to correct fine points. However, I do not feel this precludes the summary result of the paper.
Author Response
Response to Reviewer 1 Comments:Review Report (Round 2)
I encourage the authors to go-over the grammar and styling of the paper with someone to correct fine points. However, I do not feel this precludes the summary result of the paper.
Response: Thank you for the valuable comment. The whole manuscript has been checked carefully to avoid any English/grammatical mistakes..
Reviewer 2 Report
The authors responded all my concerns and the article may be accepted as it is.
Author Response
Response to Reviewer 2 Comments: Review Report (Round 2)
The authors responded all my concerns and the article may be accepted as it is.
Response: Thank you for the valuable comment.
Reviewer 3 Report
The authors have provided a careful revision and addressed all comments.
Author Response
Response to Reviewer 3 Comments: Review Report (Round 2)
The authors have provided a careful revision and addressed all comments.
Response: Thank you for the valuable comment.